

# A Local Terrain Smoothing Approach for Stabilizing Microscale and High-Resolution Mesoscale Simulations: a Case Study Using FastEddy® (v3.0) and WRF (v4.6.0)

Eloisa Raluy-López[1], Domingo Muñoz-Esparza[2], and Juan Pedro Montávez[1]

[1]Physics of the Earth, Regional Campus of International Excellence (CEIR) "Campus Mare Nostrum", University of Murcia, Murcia, Spain

[2]Research Applications Laboratory, NSF National Center for Atmospheric Research (NCAR), Boulder (CO), USA

**Correspondence:** Eloisa Raluy-López (eloisa.raluyl@um.es) and Juan Pedro Montávez (montavez@um.es)

**Abstract.** High-resolution simulations at both mesoscale and microscale increasingly rely on detailed terrain datasets, but terrain-following coordinate models can suffer from numerical instabilities in steep-slope regions. To address this issue, terrain smoothing is typically applied in numerical weather prediction models, though conventional global smoothing unnecessarily reduces resolution across the entire domain. This study presents a localized terrain smoothing approach designed to prevent numerical instabilities while preserving terrain details. Different smoothing strategies were tested for efficiency, computational cost, and terrain preservation. The final approach applies a Gaussian filter with adaptive standard deviation within a localized 3×3 grid, with a blending factor of 0.2, and treating all the steep-slope points simultaneously. Integrated into the NCAR's FastEddy® LES and WRF mesoscale community models, this technique effectively prevents terrain-driven instabilities in high-resolution simulations over complex terrain. The proposed local filtering method helps minimizing loss of terrain detail and avoiding the need for excessively strong numerical filtering during run time to stabilize the simulations. This method is computationally efficient, easy to implement, and adaptable to other models, providing a robust solution to improve numerical stability while maintaining high-resolution terrain features.

## 1 Introduction

The accurate representation of terrain is essential in atmospheric modeling. It influences wind patterns, turbulence, boundary layer development, orographic precipitation and other key meteorological processes (e.g., Raderschall et al., 2008; Geerts et al., 2011; Liang et al., 2020; Ramelli et al., 2021; Wise et al., 2022)). Many numerical models rely on terrain-following coordinate systems to capture these effects, such as the Weather Research and Forecasting model (WRF; Skamarock et al., 2021), and the FastEddy® large-eddy simulation (LES) model (Sauer and Muñoz-Esparza, 2020; Muñoz-Esparza et al., 2022). However, when working with high-resolution datasets, steep terrain slopes can introduce numerical instabilities, adversely affecting the accuracy of the simulations (Mahrer, 1984; Schär et al., 2002; Klemp et al., 2003; Zängl et al., 2004; Lundquist et al., 2010). This issue is of particular concern when critical slope thresholds of around 35° are exceeded, although some models may tolerate slightly steeper gradients.



To address numerical instabilities over steep terrain, it is common to apply terrain smoothing (e.g., Cannon et al., 2017; Skamarock et al., 2021), or to increase damping constants (Arnold et al., 2012) and time off-centering parameters for vertically propagating sound waves (see e.g., Marjanovic et al., 2014). However, increasing these values may lead to unphysical results, thus its implementation is not advised (Arnold et al., 2012). Current approaches usually involve applying global smoothing methods across the entire domain, which inevitably results in a loss of terrain detail and resolution in regions where the smoothing was not needed. Moreover, as the model resolution increases, so does the number of grid points with steep slopes, underscoring the need for alternative terrain smoothing strategies.

Some recent works tried to address the loss of terrain properties when a smoothing approach is applied. For instance, Bouëdec et al. (2025) developed a smoothing method that preserves the orographic features of the terrain. However, this method is still based on a global approach that modifies the whole domain. Sheridan et al. (2023) introduced a closest approach to localized smoothing with a targeted method. It blends a minimally smoothed terrain with a strongly smoothed one, with the latter being dominant over the localized steep-slope points. Nevertheless, this method still modifies the entire domain to some extent.

This study presents the development and implementation of a local terrain smoothing approach designed to mitigate numerical instabilities in a mesoscale model (WRF) and a microscale LES model (FastEddy®; hereafter referred to simply as FastEddy), and that is easily adaptable to other models. Various smoothing techniques were evaluated, including both simultaneous and sequential approaches. The most effective method was selected and implemented, following a thorough performance analysis, considering the number of iterations required for convergence, computational cost, and, most importantly, the degree of terrain distortion.

The structure of this paper is as follows. Section 2 describes the methodology, including the atmospheric models and high-resolution terrain datasets employed, followed by a brief overview of the local terrain smoothing techniques and the approach used to evaluate their performance. Section 3 presents the results and is divided into two main parts. The first part analyzes the performance of the different smoothing methods and identifies the most suitable one. The second part illustrates the application of the selected method to a case study where the simulation failed due to CFL-related instabilities in both WRF and FastEddy. Finally, Sect. 4 summarizes the main conclusions.

## 2 Methodology

### 2.1 Model setup and terrain representation

The city of Murcia, located in southeastern Spain, was selected as the study area. This region lies within a valley bordered by mountains to the north and south, featuring areas of complex orography. The WRF mesoscale model, version 4.6.0, and the FastEddy LES model, version 3.0, were employed in combination to analyze terrain-driven instabilities across both mesoscale and microscale domains.



### 2.1.1 WRF configuration

The WRF configuration consists of four one-way nested domains with horizontal resolutions of 9, 3, 1, and 0.2 km (Fig. A1),
approximately centered over the city of Murcia. The vertical grid includes 45 levels, with increased resolution near the surface,
and the model top set at 50 hPa.

Terrain elevation data for domains 1 to 3 were obtained from the Global Multi-resolution Terrain Elevation Data 2010
(GMTED2010; Danielson and Gesch, 2011) dataset with a spatial resolution of 30 arcsec (∼1 km), which is the default option
included in the WRF Preprocessing System (WPS). For the innermost domain (d04), covering an area of approximately $60 \times 60$
60  km with $\Delta$ = 200 m, the resolution of this default terrain dataset was found to be too coarse. Therefore, the Shuttle Radar
Topography Mission dataset with a spatial resolution of 90 m (SRTM90; Farr et al., 2007) was used instead. In this high-
resolution domain, terrain slopes locally exceed the empirical thresholds recommended for mesoscale modeling. Specifically,
Fig. 1 (a) shows the number and spatial distribution of grid points where such steep slopes of more than 30° are present. This
more conservative threshold is required to ensure numerical stability in this WRF simulation, as discussed later in Sect. 3.2.
The rest of the WRF configuration, including further surface characterization and physical parametrizations, is provided in
Appendix A.

### 2.1.2 FastEddy configuration

The FastEddy configuration consists of a $15 \times 15$ km domain approximately centered on the city of Murcia (see Fig. 1 (b)),
with a horizontal spatial resolution of 10 m ($(N_x, N_y)$ = (1536, 1530)). The vertical extent reaches up to 2700 m, divided into
90 levels using vertical stretching. This setup ensures an approximately isotropic resolution of 10 m near the surface, which
gradually increases with height. Figure 1 (b) also shows the extended FastEddy domain ($20 \times 20$ km), from which the inner
$15 \times 15$ km domain is extracted. The different terrain smoothing methods are applied over this extended domain.

The terrain elevation data were extracted from the Spanish National 2nd-Coverage Digital Terrain Model (MDT02; 2015–2021;
Instituto Geográfico Nacional, 2015), with a native grid spacing of 2 m, obtained from the download center of the National Ge-
ography Institute of Spain. Once resampled to 10 m to match the established FastEddy resolution, problematic points emerge
in areas with slopes exceeding 35° (Fig. 1 (b)). Details on land use, surface properties, the selected physical parametrizations,
and the coupling with WRF are provided in Appendix B.

### 2.1.3 Case study

To explore the occurrence and impact of terrain-driven numerical instabilities under realistic atmospheric conditions, a repre-
sentative case study was selected. The case study focuses on a warm and windy episode that occurred on 26–27 May 2023. The
simulation period for WRF spans from 12:00 UTC on 26 May to 00:00 UTC on 28 May, with an hourly output frequency. The
FastEddy simulation covers the period from 11:00 to 15:00 UTC on 27 May, with model output recorded at 5-minute intervals.

This event features persistent north-easterly winds interacting with complex terrain. As illustrated in Fig. 1, both mod-
els exhibit pronounced numerical instabilities visible in the vertical wind component—exceeding 100 m/s in the case of





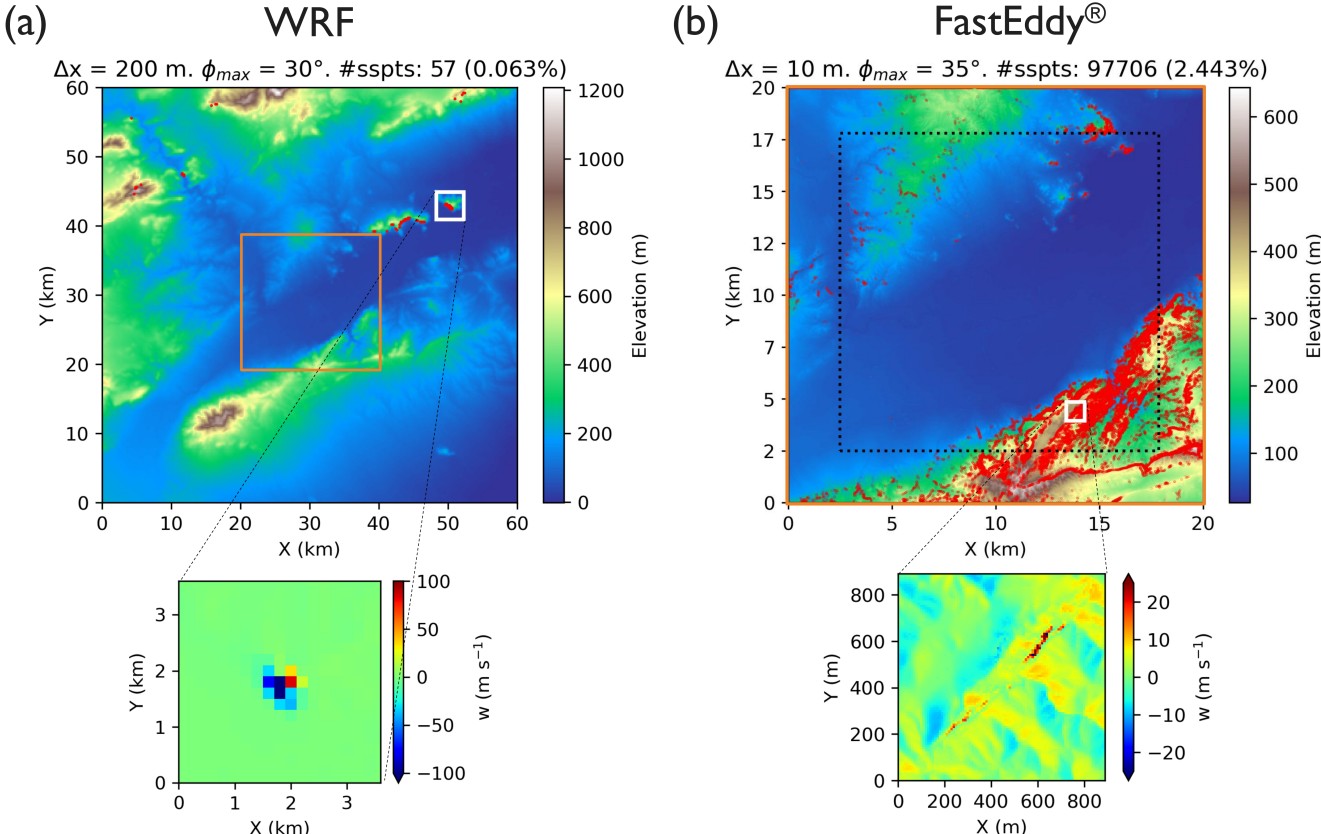

**Figure 1.** (a) Innermost WRF domain consisting on a 60×60 km area at 200 m resolution centered on the city of Murcia. The steep-slope points (sspts; slopes higher than 30°) are represented with red dots. The orange square represents the extended FastEddy domain. (b) Extended FastEddy domain of 20×20 km at 10 m resolution centered on the city of Murcia. The steep-slope points (sspts; slopes higher than 35°) are represented with red dots. The dotted black square represents the 15×15 km FastEddy domain. The white squares in both domains indicate areas of terrain-driven numerical instabilities in the simulations. The zooms to the white squares show the vertical component of the wind right before the models crush.

WRF—which ultimately lead to CFL errors and cause the simulation to crash within just a few time steps. In the case of FastEddy, these instabilities are particularly concentrated in the mountainous southeastern portion of the domain, where steep-slope points give rise to anomalous flow structures and unphysical vertical wind speeds. Note that the selected case is simply for illustration purposes, and that numerical instabilities may arise irrespective of the simulated days and/or weather conditions, since these are ultimately driven by static terrain features.





## 2.2 Local smoothing approaches

This study proposes and evaluates several terrain smoothing methodologies aimed at preventing the emergence of terrain-driven numerical instabilities, such as those illustrated in Fig. 1. All the proposed methods apply a local smoothing based on the use of a Gaussian filter over a variable-size window centered on each grid point that exceeds slope thresholds. The Gaussian filter assigns weights to surrounding cells according to a normal distribution, with higher weights near the center and decreasing weights further away. The standard deviation parameter, $\sigma$, controls the spread of this distribution: smaller $\sigma$ values result in more localized smoothing, while larger $\sigma$ values produce a broader and stronger smoothing effect. The Gaussian filter was chosen due to its effectiveness in attenuating high-frequency noise while preserving larger-scale topographic features. Furthermore, the use of a localized method ensures that the majority of the grid points are not being modified in any way, as only the steep-slope points (and their immediate surroundings) are smoothed out. Two smoothing strategies are considered: sequential, in which only the steepest point is treated at each iteration, and simultaneous, where all problematic points are treated in a single iteration. In all cases, the maximum number of iterations is set to 1.5 times the number of grid points with steep slopes.

The first step in all methods is the identification of grid points exhibiting steep slopes; that is, points where the terrain gradient in any direction exceeds a predefined threshold. This threshold was set to 35° throughout the testing phase to enable a consistent comparison between methods. Once the most suitable method is selected, the threshold remains a user-configurable parameter in the final implementation. Smoothing is then applied based on the specific characteristics of each method. The approaches evaluated in this study are summarized below.

1. Simple sequential 3×3 method. In each iteration, a Gaussian filter with an adaptive $\sigma$ is applied to a 3×3 window centered on the point of maximum slope (see the first method in Fig. 2). The process is repeated until the maximum

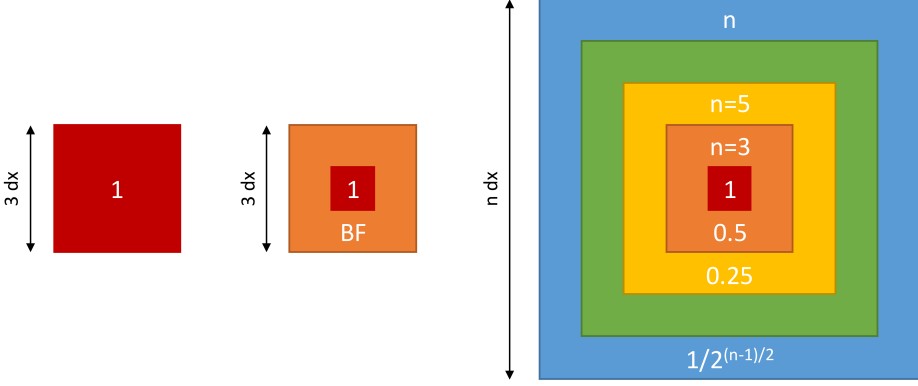

**Figure 2.** Conceptualization of the smoothing approaches. From left to right: simple 3×3 method, 3×3 method with blending, and N×N method with progressive blending.





slope does not exceed the threshold. In each iteration, the value of $\sigma$ starts at 1.0 and can be increased in steps of $\Delta\sigma =$ 1 up to $\sigma_{max} = 25$ if no steep-slope points are corrected with the previous $\sigma$ value.

2. Sequential 3×3 with blending method. Similar to Method 1, but the window's edge points' altitude is a combination of the original and smoothed terrains according to a blending factor (BF). The BF represents the proportion of modified terrain in the combination with the original to obtain the smoothed one (see the second method in Fig. 2). In this work,
BFs of 0.2, 0.5, and 0.8 were tested. Method 1 is equivalent to Method 2 with a BF of 1.

3. Sequential N×N with progressive blending method. Similar to Method 2, but the size of the window is adjustable and the altitude of each point inside it is a combination of the original and smoothed terrain according to its distance to the center. The further to the center, the smaller the proportion of the filtered terrain in the result. This study presents the N = 5, N = 7 and N = 9 configurations with a progressive BF according to Eq. (1) (see also the third method in Fig. 2).
Method 2 with a BF = 0.5 is equivalent to Method 3 with N = 3.

$$\mathrm{BF}_n = \frac{1}{2^{\frac{n-1}{2}}}, \text{with } n = 1, 3, 5, ..., \mathrm{N} \tag{1}$$

4. Simultaneous 3×3 with blending method. Same as Method 2, but all the steep-slope points are treated in the same iteration. The following iteration first recomputes the slope field. Configurations with BF of 0.05, 0.1, 0.2, 0.5 and 0.8 are presented.

5. Simultaneous N×N with progressive blending method. Same as Method 3, but all the steep-slope points are treated in the same iteration. The following iteration first recomputes the slope. Configurations with N = 7 and N = 9 are presented.

## 2.3   Performance evaluation and method selection

To identify the best method, a performance analysis was carried out to balance computational cost and terrain distortion. For comparison, the analysis also includes a global smoothing approach. This method applies a Gaussian filter over the entire
domain, following the same adaptive-$\sigma$ strategy as the local methods: each iteration begins with $\sigma = 1.0$ and increases in increments of $\Delta\sigma = 1$ up to a maximum of 25 if no steep-slope points are corrected with the previous $\sigma$ value.

The first part of the analysis focuses on a comparison of the 15 smoothing configurations in terms of computational time and number of iterations to converge, aiming to identify the most time-efficient approaches. Next, terrain distortion is evaluated as the relative elevation difference with respect to the original topography. The optimal method is then selected based on a balance
between efficiency and minimal topographic alteration. Finally, slope distributions and power spectral density are compared across the original, locally smoothed, and globally smoothed topographies.

Since the extended FastEddy domain has a higher spatial resolution than the WRF domains, it presents a greater number of steep-slope points. Therefore, the performance analysis of the different smoothing methods is conducted using this 20×20 km domain. For the sake of simplicity, the domain is upscaled to 25 m (Fig. 3), which significantly reduces the computational cost
of the analysis while preserving the validity of the conclusions. Once the most suitable method is identified, it is applied to



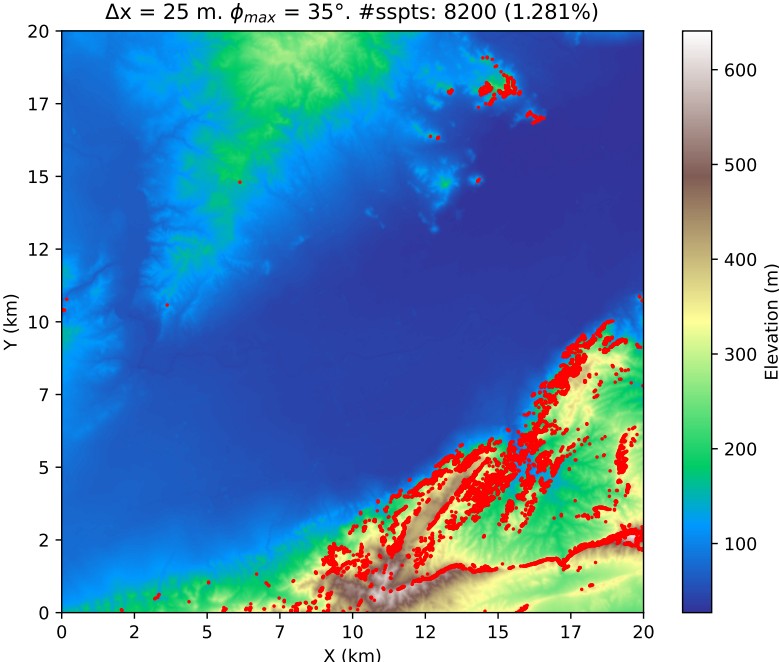

**Figure 3.** Extended FastEddy domain upscaled to 25 m resolution. The steep-slope points (sspts; points with slopes exceeding 35°) are represented with red dots.

the full-resolution FastEddy domain ($\Delta$ = 10 m) and the fourth WRF domain ($\Delta$ = 200 m). For both, computational time and induced terrain distortion are quantified, as well as the impacts on slope distribution and on the terrain power spectral density.

Finally, the originally numerically unstable simulations are re-run using the smoothed topography in order to verify whether the application of the selected method allows them to run successfully.

## 3 Results and discussion

### 3.1 Methods performance and selection

The performance of the 15 smoothing methods, including global smoothing, was evaluated for the extended FastEddy domain at 25 m resolution. First, the computational efficiency of each method was assessed in terms of the number of iterations and total time required to reach convergence. Second, the terrain distortion introduced by each method was quantified relative to the original topography. Based on both criteria, the method that best balances computational speed and terrain preservation was selected.




### 3.1.1 Time/iterations to reach convergence

Figure 4 shows the performance of the 14 tested local smoothing methods. It displays the reduction in the number of steep-slope points and the corresponding decrease in slope magnitude as a function of the number of iterations. Sequential methods require

significantly more iterations, especially the simple method and those approaches with BFs of 0.8, 0.5, and 0.2. In particular, the latter fails to reach convergence within the maximum allowed number of iterations. However, in cases with fewer problematic points, this method might still converge. The N×N methods with progressive blending require fewer iterations with larger values of N. Simultaneous methods are substantially faster, requiring up to two orders of magnitude fewer iterations. Among them, the 3×3 simultaneous methods with BF of 0.8 and 0.5 are the ones that need the fewest iterations. The number of required

iterations decreases with higher BFs; i.e., when a smaller portion of the original terrain is preserved in the boundary cells of the smoothed surface. The 3×3 method with BF = 0.05 is the simultaneous approach that needs the most iterations, followed by the N×N method with N = 9. For all methods, the majority of the iterations were performed with $\sigma = 1$, indicating that minimal smoothing was sufficient in most cases.

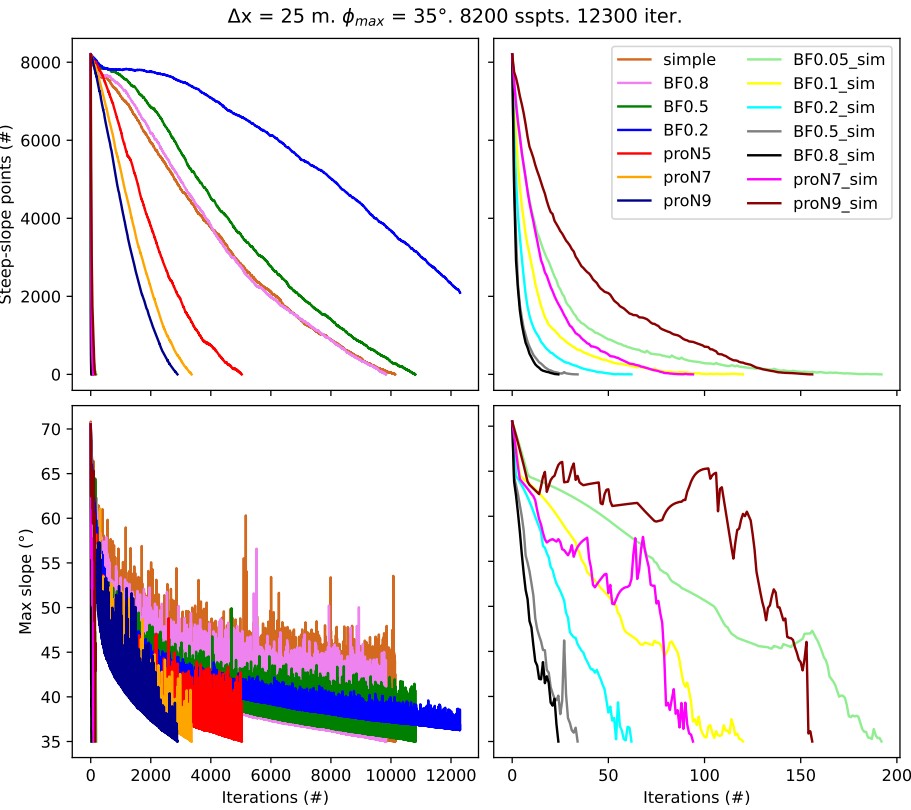

**Figure 4.** Number of steep-slope points and maximum slope as a function of the number of iterations for each local smoothing method. The panels on the right focus on the simultaneous methods, providing a zoomed-in view due to their faster convergence.





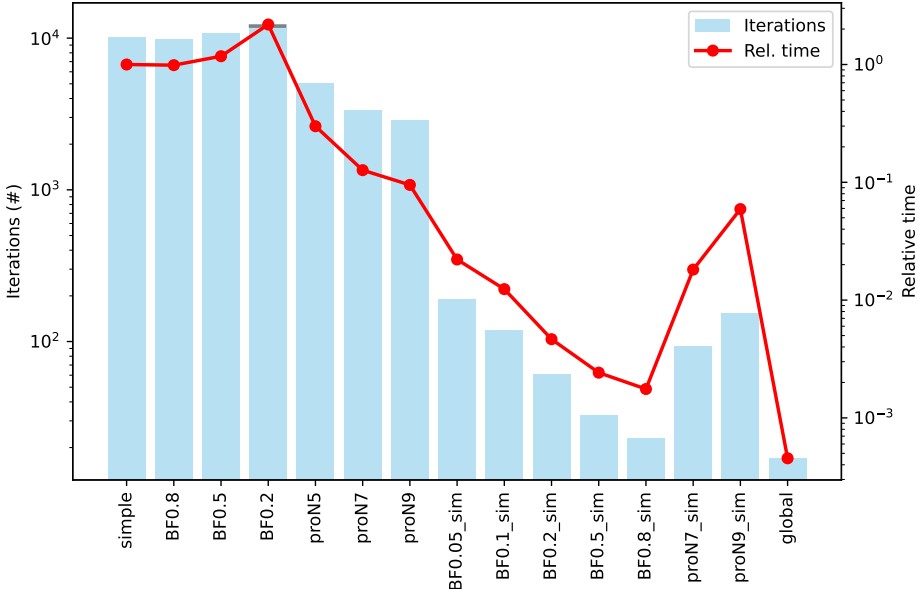

**Figure 5.** Bar plot of the number of iterations to convergence for each smoothing method. The bar capped with a grey line indicates non-convergence within the maximum iteration limit. Red dots show relative convergence time compared to the simple sequential method.

Figure 5 presents a bar plot showing the number of iterations required to reach convergence for the 14 local smoothing methods and the global method. Additionally, the computational time normalized relative to the simple sequential method is indicated in red. The global smoothing method is by far the fastest, followed by the simultaneous methods. In terms of computation time, all simultaneous methods significantly outperform the sequential methods. Specifically, for this domain with 8200 steep-slope points exceeding 35°, the simple method required 859 s to reach convergence using an Intel Xeon Gold 6140 CPU (2.30 GHz). In contrast, simultaneous methods completed in a range between 1.5 (BF0.8_sim) and 51 (proN9_sim) s. The global method required only 0.39 s to solve the problematic points in 18 iterations with $\sigma = 1$. All indicated runtimes may vary depending on the available computational resources. Based on computational time alone, any of the simultaneous methods would be suitable candidates for selection.

### 3.1.2 Terrain modification

A terrain distortion analysis was performed to evaluate the impact of the smoothing process. Figure 6 shows the percentage differences between the smoothed and original terrains. The global method introduces the highest distortion, altering areas that did not require smoothing. In contrast, the 3×3 simultaneous methods with BFs of 0.05, 0.1, and 0.2 best preserve the original terrain, with lower differences observed for smaller BFs. Those methods with BFs of 0.5 and 0.8, as well as the approaches with progressive blending (both sequential and simultaneous), introduce more substantial modifications to the original terrain. The sequential method with BF = 0.2, which did not reach convergence, is excluded from this analysis, as the reported distortions





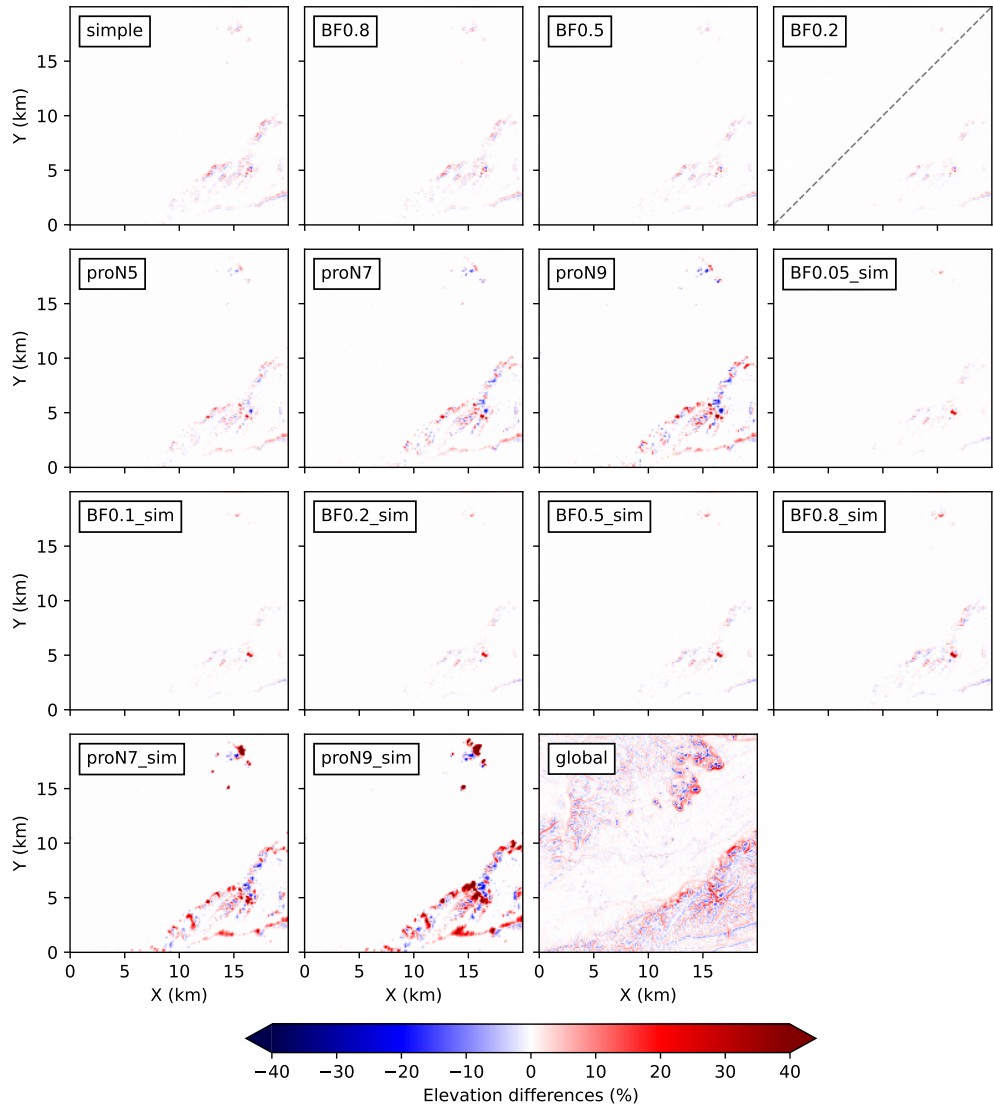

**Figure 6.** Relative elevation differences (%) between the smoothed and original terrains for all smoothing methods. For the BF0.2 method, its reported distortion correspond to non-converged results (indicated with a diagonal grey dashed line), which may differ from final greater modifications.

correspond to a non-converged state and may not reflect the full extent of terrain modification that it would produce upon convergence.



### 3.1.3 Method selection

The simultaneous $3 \times 3$ method with a BF of 0.2 is the fastest among the least terrain-altering approaches. For this reason, it was selected as the most suitable method in terms of both efficiency and terrain preservation. Figure 7 presents the slope distribution and power spectral density of the terrain smoothed using this method, alongside those of the globally smoothed and original terrains (see panels with $\Delta = 25$ m). The local method effectively limits slopes to the defined threshold, while the global method

**Figure 7.** Slope density distribution (left) and power spectral density (right) of the original (red), globally smoothed (green), and locally smoothed (blue) terrains of the different FastEddy and WRF domains. From top to bottom: fourth WRF domain ($\Delta = 200$ m), upscaled extended FastEddy domain ($\Delta = 25$ m), extended FastEddy domain ($\Delta = 10$ m). Grey dashed lines indicate the maximum slope thresholds, established at 30° for the mesoscale and at 35° for the microscale.





produces a smoother terrain, yet some maximum slopes remain close to the threshold value. Additionally, the local smoothing preserves the power spectrum of the original terrain to a large extent, closely matching it at both large and small wavelengths. A slight reduction of power is observed between $10^{-3}$ and $10^{-2}$ m$^{-1}$, corresponding to intermediate terrain features. A modest

increase in power is observed for wavenumbers greater than $10^{-2}$ m$^{-1}$, likely reflecting small-scale noise introduced during the blending process. In contrast, the global method strongly suppresses spectral power at wavenumbers larger than $3 \times 10^{-4}$ m$^{-1}$, removing all medium- and fine-scale terrain features. A comparison of the terrain distortion introduced by the global and local smoothing approaches is shown in Fig. 8. In comparison to the global approach, the local smoothing technique preserves the terrain features to a much greater extent, introducing only minor, localized distortions.

## 195 3.2 Model implementation and case study: preventing terrain-induced numerical instabilities

Once the optimal smoothing method was selected, it was implemented in both the WRF and FastEddy modeling systems. For WRF, the algorithm was integrated as an additional step within the WPS. After generating the geographical files using the *geogrid* utility, the smoothing algorithm is applied to produce a new set of files with locally smoothed topography, with all other fields preserved. The subsequent steps of the WPS workflow remain unchanged. The algorithm was implemented in

Python using standard libraries. The Gaussian filter used for smoothing is imported from SciPy (Virtanen et al., 2020). The source code, along with usage instructions and an example file, is openly available on Zenodo (Raluy-López et al., 2025b). The maximum slope threshold can be adjusted by the user; however, a value around 30° is recommended to ensure stability in high-resolution WRF simulations. While the present analysis focuses on mesoscale WRF applications, the developed smoothing technique is fully compatible with WRF-LES setups as well.

When applied to the innermost WRF domain with a horizontal resolution of $\Delta$ = 200 m and an imposed maximum slope of 30°, the smoothing procedure completed in under one second, correcting 57 steep-slope points in just five iterations. Figure 7 (see panels with $\Delta$ = 200 m) shows the slope distribution and power spectral density of the original and smoothed terrains. As previously seen for the upscaled FastEddy domain, the global smoothing method (2 iterations with $\sigma$ = 1) modifies the terrain to a large extent, eliminating most fine-scale details despite the presence of only a limited number of steep-slope points.

Figure 8 displays the original terrain of this WRF domain alongside the globally and locally smoothed versions, as well as the corresponding elevation differences between the local smoothing results and the original topography. Notably, the terrain distortion introduced by the local smoothing method is minimal, particularly when compared to the substantial differences produced by the global smoothing approach.

Under the meteorological conditions described in Sect. 2.1.3, the simulation would crash after only a few timesteps due to

215 numerical instabilities that ultimately led to CFL errors. Without any terrain smoothing, the simulation could run successfully by applying high values of the time off-centering parameter for vertically propagating sound waves (*epssm* $\geq$ 0.9). However, as previously discussed in the Introduction, increasing this parameter may result in unphysical behavior and is therefore not recommended (Arnold et al., 2012). High-resolution simulations aim to maximize accuracy, making such adjustments undesirable. With the implementation of the local smoothing method, the simulation successfully completes without terrain-induced

numerical instabilities, requiring the modification of only a few points and resulting in only a modest increase in computa-



**Figure 8.** Comparison of original and smoothed topographies across the three modeling domains. Each row represents a different domain. From top to bottom: the fourth WRF domain ($\Delta = 200$ m), the upscaled extended FastEddy domain ($\Delta = 25$ m), and the extended FastEddy domain ($\Delta = 10$ m). Columns show, from left to right: the original topography, the terrain after global smoothing, the terrain after local smoothing, and the percentage elevation differences between the locally smoothed and the original topographies.

tional cost. In this case, the stricter 30° slope threshold was necessary, as the simulation failed when using a 35° threshold, underscoring a greater model sensitivity to steep terrain.





In the case of FastEddy, the selected local smoothing method has been fully integrated into the model source code. The latest publicly available version of the model (v3.0; National Center for Atmospheric Research (NCAR), 2025) already includes this improvement, replacing the previously used global smoothing approach. The algorithm is executed during the pre-processing *SimGrid* step, which performs the generation of the domain grid files including all geophysical variables. As an illustrative example, smoothing the extended 20×20 km FastEddy domain at 10 m takes less than three minutes, resolving 97,706 steep-slope points in 304 iterations using a single Intel Xeon Gold 6140 CPU (2.30 GHz). The slope distributions and power spectral densities of the terrain before and after smoothing are shown in Fig. 7. Figure 8 (see panels with $\Delta = 10$ m) compares the original terrain of the extended domain with the globally smoothed version (110 iterations with $\sigma = 1$) and the locally smoothed version, also showing the elevation differences introduced by the local method. The conclusions drawn for the 10 m resolution extended domain are consistent with those obtained from the upscaled analysis. Following the application of the local smoothing algorithm, the previously failing simulation runs to completion without any terrain-induced numerical instabilities and without the appearance of spurious values or structures.

Although the global smoothing method also ensures numerical stability in both models, it substantially compromises fine-scale terrain features. These high-resolution details are essential for accurately representing local processes, particularly in simulations aimed at capturing small-scale atmospheric dynamics. The local smoothing method offers improved preservation of the original fine-scale topography, allowing for more realistic terrain representation without sacrificing numerical stability. In addition, its local nature makes it suitable to also remove noise or other artifacts from terrain datasets that would similarly lead to numerical instabilities.

## 4 Conclusions

High-resolution atmospheric modeling over complex terrain poses significant challenges due to the presence of steep slopes, which often lead to numerical instabilities and failed simulations. These problems are particularly relevant in models that employ terrain-following coordinate systems, such as WRF or FastEddy, where the accuracy of the simulations is tightly linked to the underlying topography. To avoid these instabilities, global terrain smoothing is frequently applied, but this solution inevitably compromises the representation of fine-scale terrain features that are essential for studying local meteorological processes. As model resolutions continue to increase and finer details become more relevant, there is a growing need for alternative approaches that ensure numerical stability while preserving the physical properties of the terrain.

In this work, a local terrain smoothing approach is developed and implemented. It is specifically designed to address numerical instabilities caused by steep slopes, while minimizing the loss of terrain detail. Several local smoothing strategies were tested and evaluated according to their performance in reducing steep-slope points, preserving topographic features, and maintaining a low computational cost. The best-performing method was the simultaneous 3×3 method with a blending factor of 0.2. It treats all the problematic points in the same iteration, applying a Gaussian filter with adaptive standard deviation over 3×3 cells. The central point is completely smoothed, while the edge cells preserve an 80 % of the original terrain information.



The selected local method was integrated into both WRF and FastEddy modeling frameworks, and its effectiveness was demonstrated in high-resolution domains. In WRF, it was added as an extra step within the WPS. In FastEddy, it replaced the previously implemented global smoother and is already included in the latest publicly available version of the model. Unlike traditional global smoothing methods, the local approach selectively modifies only problematic areas, allowing the rest of the terrain to remain unaltered and ensuring a more physically realistic representation of complex topography. Furthermore, the

increase in preprocessing time was negligible, with even complex domains being smoothed in just a few minutes using standard hardware.

    The application of the selected method successfully prevents terrain-driven instabilities, without the need for artificial damping or large-scale topographic simplifications. When applied to a complex terrain, prone to triggering numerical instabilities, it allows the previously failing simulations to complete without errors.

In conclusion, the developed local smoothing method ensures numerical stability while preserving the essential fine-scale structure of the original terrain. This makes it particularly well suited for applications that demand high spatial fidelity. This method has been successfully implemented in both a mesoscale and a microscale model, and its design allows for straightforward adaptation to other modeling frameworks.



**Appendix A: WRF setup details**

Across all domains, land cover information was derived from the CGLC-MODIS-LCZ dataset at 100 m resolution (Demuzere et al., 2023), and soil texture classification was obtained from the HWSD version 2-based global dataset tailored for WRF (Bonafè et al., 2024), available at 30 arcsec (∼1 km) resolution. All other geophysical datasets required by the WPS were used as provided by default.

    The initial and boundary conditions were derived from the ERA5 reanalysis, with a spatial resolution of approximately 27
275  km, and updated in the model every 3 h. This WRF configuration follows the CONUS physics suite, including: the Thompson microphysics scheme (Thompson et al., 2008); the Tiedtke cumulus parametrization (Zhang et al., 2011), activated only in the outermost domain, as domains 2 to 4 operate at convection-permitting scales; the Rapid Radiative Transfer Model for General circulation models for both shortwave and longwave radiation (RRTMG; Iacono et al., 2008); the Mellor-Yamada-Janjic boundary layer scheme (MYJ; Janjić, 1994); the Noah land surface model (Chen and Dudhia, 2001); and the Eta similarity
scheme for surface layer processes (Janjić, 1994). The Building Environment Parameterization (BEP; Martilli et al., 2002) urban canopy scheme is also activated, along with the use of the local climate zones (LCZs).

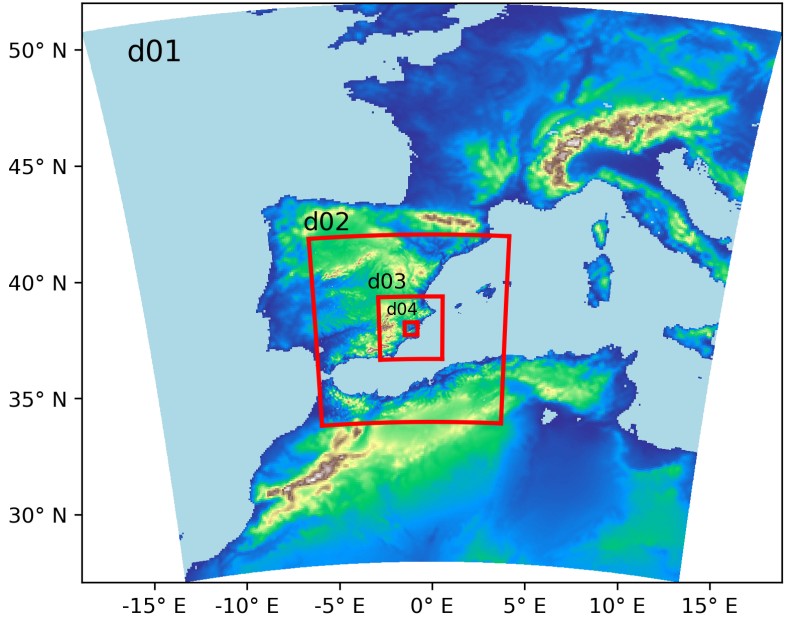

**Figure A1.** WRF study area, consisting in four one-way nested domains of 9, 3, 1, and 0.2 km approximately centered in the city of Murcia (Spain).



**Appendix B: FastEddy setup details**

Land cover information was derived from the 2014 SIOSE database, a high-resolution national land-use inventory with a minimum mapping unit of 1 ha in urban areas, 2 ha in rural areas, and 0.5 ha in water bodies (Instituto Geográfico Nacional, 2014). This dataset was translated into the MODIS-compatible land-use categories used by WRF (including the LCZs) using a reclassification table developed in this work and specifically adapted to the urban characteristics of the city of Murcia (Table B1). The surface roughness length ($z_{0m}$) for each land-use category was assigned according to the new proposed values in Table B2, based on the WRF LANDUSE.TBL and a supporting literature review. The effect of buildings was not explicitly included in these FastEddy simulations.

FastEddy was forced using the output from the WRF simulations, specifically from domain 3 ($\Delta$ = 1 km), including all relevant prognostic variables. The coupling between WRF and FastEddy was performed in an offline, one-way manner. In this configuration, both two-dimensional (e.g., skin temperature and surface water vapor) and three-dimensional (e.g., wind components, temperature, and humidity) fields are extracted from the WRF simulation and interpolated onto the boundaries of the FastEddy domain. These interpolated fields are then applied as time-dependent Dirichlet boundary conditions along the lateral, top, and bottom boundaries. At the surface, time-evolving maps of skin temperature and specific humidity are imposed. The boundary conditions are updated every 5 minutes. A model time step of 0.02 s was selected to meet the numerical stability requirements associated with resolving acoustic wave propagation.

The physical configuration of FastEddy employs a prognostic subgrid-scale turbulence scheme based on the 1.5-order turbulence kinetic energy closure originally proposed by Deardorff (1980), and further refined by Lilly (1966). Surface fluxes of momentum, sensible heat, and latent heat are computed using local surface-layer gradients according to the Monin–Obukhov similarity theory (Monin and Obukhov, 1954), ensuring physically consistent exchange processes at the land–atmosphere interface. A saturation adjustment scheme is included to represent condensation and evaporation processes, with non-precipitating clouds. Prognostic equations describe air density, the three velocity components, potential temperature, and water vapor mixing ratio. These are discretized using a fifth-order upwind advection scheme and integrated in time using a third-order Runge–Kutta method to ensure numerical stability and accuracy. To promote the rapid development of realistic turbulent structures within the domain, which is initially forced with turbulence-free WRF boundary conditions, the cell perturbation method is applied (Muñoz-Esparza et al., 2014, 2015; Muñoz-Esparza and Kosović, 2018). This method efficiently triggers and sustains turbulence near the inflow boundaries. Finally, surface thermal roughness length ($z_{0t}$) is dynamically computed using the Zilitinkevich parametrization (Zilitinkevich, 1995).





**Table B1.** Reclassification from SIOSE (CODIIGE) to Modified MODIS IGBP Noah land-use categories without lakes, including the LCZs. The symbol "*" indicates a specific reclassification for Murcia, may vary from city to city.

| ID | SIOSE/CODIIGE classification | MODIS classification with LCZs | ID |
|---|---|---|---|
| 111 | Downtown urban area | Compact midrise * | 52 |
| 112 | Expansion urban area | Compact lowrise * | 53 |
| 113 | Discontinuous urban area | Open lowrise | 56 |
| 114 | Green urban area | Cropland/natural vegetation mosaic | 14 |
| 121 | Agricultural and/or livestock facility | Sparsely built | 59 |
| 122 | Forestry facility | Sparsely built | 59 |
| 123 | Mineral extraction site | Bare rock or paved | 61 |
| 130 | Industrial | Large lowrise * | 58 |
| 140 | Service infrastructure | Open midrise | 55 |
| 150 | Agricultural settlements and cropfields | Croplands | 12 |
| 161 | Road and rail networks | Bare rock or paved | 61 |
| 162 | Port | Large lowrise | 58 |
| 163 | Airport | Bare rock or paved | 61 |
| 171-172 | Supply infrastructure, waste infrastructure | Heavy industry | 60 |
| 210 | Herbaceous crops | Grasslands | 10 |
| 220 | Greenhouse | Sparsely built | 59 |
| 231 | Citrus fruit trees | Evergreen broadleaf forest | 2 |
| 232 | Non-citrus fruit trees | Deciduous broadleaf forest | 4 |
| 233 | Vineyard | Closed shrublands | 6 |
| 234 | Olive grove | Evergreen broadleaf forest | 2 |
| 235-236 | Other woody crops, combination of woody crops | Open shrublands | 7 |
| 240 | Pasture | Grasslands | 10 |
| 250 | Mixed crops | Croplands | 12 |
| 260 | Crops and natural vegetation | Cropland/natural vegetation mosaic | 14 |
| 311 | Broad-leaved forest | Deciduous broadleaf forest | 4 |
| 312 | Coniferous forest | Evergreen needleleaf forest | 1 |
| 313 | Mixed forest | Mixed forest | 5 |
| 320 | Grasslands | Grasslands | 10 |
| 330, 340 | Shrublands, mixed vegetation | Open shrubland | 7 |
| 351 | Beaches/dunes/sands | Barren or sparsely vegetated | 16 |
| 352 | Bare rocks | Bare rock or paved | 61 |
| 353-354 | Burnt, bare soil | Barren or sparsely vegetated | 16 |
| 411-414 | Inland marshes, peat bogs, salt marshes, salines | Permanent wetlands | 11 |
| 511-515 | Water course, lake or lagoon, water reservoir, artificial water surface, sea | Water | 17 |
| 516 | Glacier and perpetual snow | Snow and ice | 15 |



**Table B2.** Roughness length values ($z_0$) of the MODIS land cover categories including the LCZs according to the LANDUSE.TBL of WRF and our new proposal.

| Category | Description | $z_{0,\text{WRF}}$ (m) | $z_{0,\text{new}}$ (m) |
|---|---|---|---|
| 1 | Evergreen needleleaf forest | 0.50 | 0.67 |
| 2 | Evergreen broadleaf forest | 0.50 | 0.72 |
| 3 | Deciduous needleleaf forest | 0.50 | 0.60 |
| 4 | Deciduoud broadleaf forest | 0.50 | 0.65 |
| 5 | Mixed forests | 0.35 | 0.66 |
| 6 | Closed shrublands | 0.03 | 0.10 |
| 7 | Open shrublands | 0.035 | 0.12 |
| 8 | Woody savannas | 0.03 | 0.06 |
| 9 | Savannas | 0.15 | 0.08 |
| 10 | Grasslands | 0.11 | 0.05 |
| 11 | Permanent wetlands | 0.30 | 0.25 |
| 12 | Croplands | 0.10 | 0.04 |
| 13 | Urban and built-up | 0.80 | 0.35 |
| 14 | Cropland/natural vegetation mosaic | 0.095 | 0.035 |
| 15 | Snow and ice | 0.001 | 0.001 |
| 16 | Barren or sparsely vegetated | 0.01 | 0.02 |
| 17 | Water | 0.0001 | 0.0005 |
| 18 | Wooded tundra | 0.30 | 0.30 |
| 19 | Mixed tundra | 0.15 | 0.15 |
| 20 | Barren tundra | 0.075 | 0.075 |
| 21-30 | Unassigned | 0.80 | 0.20 |
| 31 | Low intensity residential | 0.80 | 0.30 |
| 32 | High intensity residential | 0.80 | 0.70 |
| 33 | Industrial or commercial | 0.80 | 0.40 |
| 34-50 | Unassigned | 0.80 | 0.20 |
| 51 | LCZ1 - Compact highrise | 0.80 | 2.00 |
| 52 | LCZ2 - Compact midrise | 0.80 | 0.75 |
| 53 | LCZ3 - Compact lowrise | 0.80 | 0.50 |
| 54 | LCZ4 - Open highrise | 0.80 | 1.50 |
| 55 | LCZ5 - Open midrise | 0.80 | 0.40 |
| 56 | LCZ6 - Open lowrise | 0.80 | 0.35 |
| 57 | LCZ7 - Lightweight lowrise | 0.80 | 0.20 |
| 58 | LCZ8 - Large lowrise | 0.80 | 0.25 |
| 59 | LCZ9 - Sparsely built | 0.80 | 0.30 |
| 60 | LCZ10 - Heavy industry | 0.80 | 0.40 |
| 61 | LCZ11 - Bare rock or paved | 0.80 | 0.004 |



*Code and data availability.* The code for the different smoothing approaches—both global and local, including the selected method—is openly available in a Zenodo repository (https://doi.org/10.5281/zenodo.16635511; Raluy-López et al., 2025a), along with the high-resolution terrain files used by FastEddy and WRF. The repository also includes the preprocessing of the terrain data and the application of the smoothing methods required to reproduce the results presented in this paper. The developed terrain smoothing algorithm is also integrated into the FastEddy modeling system, included in the latest official release of FastEddy, which is publicly available via its GitHub repository (v3.0; National Center for Atmospheric Research (NCAR), 2025). A version adapted for the WRF model is likewise openly available on Zenodo (https://doi.org/10.5281/zenodo.16265023; Raluy-López et al., 2025b), together with detailed usage instructions and a sample NetCDF terrain file.

*Author contributions.* ERL developed the local terrain smoothing algorithms and performed the analyses presented in this paper. DME and JPM contributed to the conceptual design of the study and supported the analysis, providing critical scientific guidance throughout the development and implementation of the algorithm. ERL led the manuscript writing, with all authors contributing to the review and editing of the final text.

*Competing interests.* The authors declare that they have no conflict of interest.

*Acknowledgements.* ERL and JPM acknowledge the ARUBA project (grant no. PID2023-149080OB-I00/MCIN/AEI/10.13039/501100011033, Ministerio de Ciencia e Innovación/Agencia Estatal de Investigación, Spain & FEDER, EU). Moreover, ERL thanks her predoctoral contract FPU (FPU21/02464) to the Ministerio de Universidades of Spain. The authors are thankful to Dr. Jeremy Sauer who implemented the original version of the global terrain smoothing filter in FastEddy.





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
