# Peer review of "A Local Terrain Smoothing Approach for Stabilizing Microscale and High-Resolution Mesoscale Simulations: a Case Study Using FastEddy® (v3.0) and WRF (v4.6.0)"

_EGUsphere, 2025_

## Author Comment (AC1)

**Authors' Response**

We would like to once again express our gratitude to the referee for their valuable suggestions and insightful comments. Their thoughtful review of our work has greatly contributed to the substantial improvement of the manuscript. Referee comments are included in **Black**. Our detailed responses are highlighted in **Blue**. The changes implemented in the revised manuscript are indicated in **Red**.

**Response to Referee #1**

line 21: the 35 degree as threshold value needs some more context, since it is heavily relied upon throughout the paper. What mechanism causes the instability at 35 degrees, and why can some models tolerate steeper gradients?

Thank you for the comment. We agree that a 35° threshold is too specific for the introduction. In fact, this is an empirical value that depends on several factors. Numerical instabilities can arise regardless of the simulated period or weather conditions, since they are ultimately linked to static terrain features that induce strong flow gradients. These instabilities are related to the model's limited ability to numerically resolve steep orography on a discretized grid, which can produce unrealistically large pressure-gradient and vertical-velocity errors. These effects are further modulated by grid spacing and time-integration settings, which explains why some model configurations can tolerate steeper slopes than others. However, atmospheric conditions can also influence their onset. As explained in the manuscript, slope thresholds of 30° and 35° were required for our mesoscale and microscale simulations to complete successfully, respectively. Nevertheless, this threshold should be adapted for each simulated case, selecting the highest value that ensures numerical stability, thus preserving the terrain as faithfully as possible.

We have modified this paragraph, providing a general approach to the problem and specifying that these are empirical limits: "This issue is of particular concern when terrain slopes exceed **empirical critical values of around 30–40°. In such cases, numerical instabilities can primarily arise from the interaction between steep orography and terrain-following coordinates at finite grid resolution, which amplifies pressure-gradient discretization errors and spurious vertical velocities (e.g., Schär et al., 2002; Klemp et al., 2003). The effective stability also depends on local resolution and time-integration settings (e.g., Courant number), so tolerable slopes may vary across models and configurations."**

line 32: what is a 'closest approach'?

Thank you for the question. We meant "a method that more closely approximates to localized smoothing while remaining global."

To avoid ambiguity, we have revised the sentence to: "**Sheridan et al. (2023) proposed a targeted method that represents a closer approximation to localized smoothing.**"

line 38: 'thorough' - I do not disagree, but adjectives like this do not belong in scientific text.

Thank you for the comment. We have removed that word.

That sentence now reads: "The most effective method was selected and implemented, following a  performance analysis [...]".

line 71: The grid spacing, probably not the resolution, increases with height.

Thank you for the comment. We have made it clearer in the text.

The full sentence now reads as follows: "This setup ensures an approximately isotropic **grid spacing** of 10 m near the surface, **with the vertical grid gradually stretching with height**."

Section 2.1.2: I find the inner and extended FastEddy domains a bit confusing: it seems that only results from the extended domain are presented, so what's the use of the inner domain in this paper? Also, the naming implies that these are two (nested) simulations (like for WRF), which is not the case, if I understand it correctly.

Thank you for the comment. FastEddy does not use nested dynamical simulations here. We define two *domains* for different purposes: an inner domain (15×15 km) where the LES is actually integrated, and an extended domain (20×20 km) used only at the preprocessing stage (*SimGrid*) to derive the geophysical inputs (terrain elevation, land use, roughness length) needed by the inner domain. The smoothing algorithms are applied and compared over the extended domain because it contains more steep-slope points and thus represents a more demanding test for terrain-related instabilities. The simulations themselves are run only in the inner domain. We have revised the text to clarify this separation.

We have included some clarifications in Section 2.1.2: "The FastEddy **simulations are performed over** a 15×15 km **inner (simulation)** domain approximately centered on the city of Murcia (see Fig. 1 (b)), with a horizontal spatial resolution of 10 m (($N_x$, $N_y$) = (1536, 1530)). The vertical extent reaches up to 2700 m, divided into 90 levels using vertical stretching. This setup ensures an approximately isotropic resolution of 10 m near the surface, with the vertical grid spacing gradually increasing with height. Figure 1 (b) also

shows the 20×20 km extended **(preprocessing)** FastEddy domain **used only in the *SimGrid* step to generate the geophysical fields (terrain elevation, land use, roughness length) for the inner domain.** The different terrain-smoothing methods are applied **and compared** over this extended domain **because it contains a larger fraction of steep-slope points, providing a more demanding test. The LES integration is performed exclusively in the inner domain to assess whether the smoothing yields numerically stable simulations. Nevertheless, any smaller domain extracted from the smoothed extended domain is expected to run without terrain-driven numerical instabilities.** ˮ

Figure 1 caption: Typo (crush instead of crash).

Thank you. We have corrected the typo.

The last sentence of the caption now reads as follows: "The zooms to the white squares show the vertical component of the wind right before the models **crash**".

line 139: How exactly is the terrain upscaled? Since the paper is about terrain data processing, this is an important detail.

The upscaling is performed employing the *block_reduce* function of the *skimage.measure* Python module. This function is used to extract the mean value inside each 25x25 m block of the original terrain. This upscaling is applied only to the testing section, for the sake of simplicity. Once we have selected the final method, the smoothing was directly applied to the original resolutions in Section 3.2.

We have included a short explanation of how the upscaling was performed: "For the sake of simplicity, the domain is upscaled to 25 m (Fig. 3) **by block-averaging over non-overlapping 25×25 m tiles**, which significantly reduces the computational cost of the analysis while preserving the validity of the conclusions."

line 149: it should be more explicitly stated what convergence entails.

Thank you for the comment. In this context, "reach convergence" means reaching the point where the smoothed result has no steep-slope points anymore, so the maximum slope falls below the imposed threshold.

We have added a brief clarification of how *convergence* is used in our analysis at its first occurrence (Section 2.3): "The first part of the analysis focuses on a comparison of the 15 smoothing configurations in terms of computational time and number of iterations to convergence **(i.e., maximum slope below the prescribed threshold, no remaining steep-slope points)**, aiming to identify the most time-efficient approaches. Next, […]"

Figure 6: What is the purpose of the metric relative elevation differences? A certain elevation difference caused by the smoothing will have a relative elevation difference that depends on the location in the domain. Why not present absolute elevation difference or slope difference?

Thank you for this question. Our goal was to compare smoothing impacts across areas with very different topographic scales. Because the lowest points in the domain are close to sea level, the main reason behind this decision was to prevent absolute elevation differences (dominated by mountainous regions) from potentially masking changes of similar relative magnitude in lower-lying areas. The relative elevation difference provides a scale-aware, dimensionless measure. We agree that absolute differences and slope differences can be more informative in some cases. We have generated an absolute-difference version of Fig. 6 (see figure below). It highlights mountainous points slightly more strongly (although they are within reasonable relative differences), but the spatial patterns and conclusions remain consistent with those from the relative metric. We therefore keep the original relative-difference version.

[Figure]

Figure 7: In the slope density distribution, it is not clear what each color means.

Thank you for the comment. We have removed excessive transparency in Fig. 7, included KDE lines for more clarity, and added y-axis units (see response to the next question).

Figure 7: All three resolutions have the same maximum wavenumber: $2 \times 10^{-2}$ m$^{-1}$, which corresponds to 50 m. Since the three panels use different resolutions, it is not clear what is meant with the wavenumber. Also, the y axes lack units.

Thank you for the comment. You are right. Due to a typo, the three spectra were computed with the same grid spacing ($\Delta = 25$ m), which made them share the same maximum wavenumber. We have corrected this, as you can see in the figure below. Consequently, all three spectra now end according to their Nyquist limit ($k_{Ny} = 1/(2\Delta)$). Therefore, with $\Delta = 25$ m, $k_{Ny} = 0.02$ m$^{-1}$, and the spectrum must end there for this resolution.

Regarding the y-axis units, the original spectra were reported in arbitrary units. We have now recomputed them as normalized 1-D power spectral densities $S(k)$ with physical units of m$^3$, such that the integral of $S(k)$ equals the variance of elevation (m$^2$). The revised figure leads to the same conclusions, and only minor wording changes were needed in the accompanying text.

We have modified the text in Section 3.1.3, according to the new figure: "A slight reduction of power is observed between  **$10^{-4}$** and $10^{-2}$ m$^{-1}$, corresponding to intermediate**-to-large** terrain features."

[Figure]

line 265: 'the developed local smoothing method ensures numerical stability', this conclusion is a bit too strong (with one case study).

Thank you for your comment. We have corrected it.

That sentence now reads: "the developed local smoothing method **helps ensure** numerical stability".